# Disarming Strategic Text: Span-Aware Counterfactuals for Robust Content Moderation

**Hardik Meisheri**[*]
Microsoft AI

**Muhammad Zaid Hassan** [†]
Manipal Institute of Technology

**Swati Tiwari**
Microsoft AI

**Puneet Mangla**
Microsoft AI

**Samarth Bharadwaj**
Microsoft AI

**Karthik Sankaranarayanan**
Microsoft AI

**Amit Singh**
Microsoft AI

## Abstract

Machine learning systems deployed in the wild must operate reliably despite unreliable inputs, whether arising from distribution shifts, adversarial manipulation, or strategic behavior by users. Content moderation is a prime example: violators deliberately exploit euphemisms, obfuscations, or benign co-occurrence patterns to evade detection, creating unreliable supervision signals for classifiers. We present a span-aware augmentation framework that generates high-quality counterfactual hard negatives to improve robustness under such conditions. Our pipeline combines (i) multi-LLM agreement to extract causal violation spans, (ii) policy-guided rewrites of those spans into compliant alternatives, and (iii) validation via re-inference to ensure only genuine label-flipping counterfactuals are retained. Across real-world ad moderation and toxic comment datasets, this approach consistently reduces spurious correlations and improves robustness to adversarial triggers, with PRAUC gains of up to +6.3 points. We further show that augmentation benefits peak at task-dependent ratios, underscoring the importance of balance in reliable learning. These findings highlight span-aware counterfactual augmentation as a practical path toward reliable ML from strategically manipulated and unreliable text data.

## 1 Introduction

Automated content moderation systems often fail when users strategically manipulate language with euphemisms, misspellings, or context-dependent phrasing. This makes moderation a prime example of unreliable data: models are asked to operate under distribution shift and deliberate adversarial tactics.

A central challenge is distinguishing true violation signals from spurious correlations. Classifiers trained on imbalanced or noisy datasets often overfit to frequent co-occurrences (e.g., benign marketing terms) instead of the causal violation spans. As a result, they are vulnerable both to false positives and to evasions by adversarial rephrasing.

Prior augmentation techniques (synonym replacement, back-translation) increase surface diversity but fail to flip labels, they rarely edit the causal spans responsible for violations. Consequently, they do not yield the hard negatives required to improve robustness.

We introduce a span-aware counterfactual augmentation framework that addresses this gap. Our method identifies violation-causing spans with high precision, rewrites them into compliant alterna-

---

[*]Corresponding author: `hmeisheri@microsoft.com`

[†]work done during internship at Microsoft AI

39th Conference on Neural Information Processing Systems (NeurIPS 2025) Workshop: Reliable ML from Unreliable Data.

tives, and validates that the label flips. This produces realistic counterfactuals that force classifiers to rely on genuine causal signals.

Our contributions are:

- A high-precision span extraction method using multi-LLM agreement (61.2% joint span agreement) that identifies causal violation spans with minimal false positives
- A policy-guided rewriting approach achieving 94.3% label-flip success while preserving context and fluency
- Empirical validation showing 6.3 percentage point PRAUC improvements with only 5-20% augmentation, compared to 31% degradation from naive masking

By grounding augmentation in span-level causal edits, our framework offers a principled approach to robustness under adversarial manipulation, with broader relevance beyond moderation.

## 2   Related Work

Automated content moderation has rapidly evolved from simple keyword-based systems to sophisticated neural approaches to combat increasingly nuanced policy violations and adversarial tactics. While modern methods, particularly those leveraging pre-trained language models, have significantly improved performance, they often struggle with spurious correlations, where models learn to associate non-causal features with policy violations. Our work addresses this challenge by focusing on three key research areas:

**Span-level text processing:** Span identification has emerged as a crucial step in moderation and broader NLP, enabling models to locate and reason over the specific text segments responsible for a label. Early work in Named Entity Recognition (NER), event detection, and extractive Question-Answering provided foundational tools for span supervision. More recent approaches explicitly incorporate label knowledge or semantic fusion modules for higher precision Yang et al. [2021]. PeerDA Xu et al. [2023] highlights the value of peer relational cues for span identification in augmentation pipelines. Beyond moderation, the explainability literature has emphasized span-level rationales as causal signals, including attention-as-explanation critiques Jain and Wallace [2019], benchmarked rationale extraction DeYoung et al. [2020] and causal rationales for robust classification Jhamtani and Clark [2020]. These works collectively underscore the importance of high-precision spans, but few integrate them into augmentation frameworks for moderation.

**Data augmentation and counterfactual generation:** Classical augmentation strategies, such as synonym replacement, random token swaps, and backtranslation Sennrich et al. [2016], Edunov et al. [2018], improve lexical diversity, but typically preserve the original causal span and label. Although useful for regularization, they rarely produce the *label-flipping hard negatives* needed to combat spurious correlations. Recent work has leveraged generative models such as BART Lewis et al. [2019] or model-in-the-loop frameworks to create synthetic data, but these remain largely label-preserving unless explicitly counterfactual. Counterfactual data augmentation Kaushik et al. [2021], Gardner et al. [2020] and controlled counterfactual generation methods such as Polyjuice Wu et al. [2021] illustrate the potential of targeted edits for robustness. However, most of these approaches operate at the sentence level rather than isolating and rewriting causal spans, limiting their applicability to fine-grained moderation.

**Hard negative mining.** Selecting examples that challenge decision boundaries has long been a central element in contrastive learning and information retrieval. Classical retrieval-based approaches include Dense Passage Retrieval Karpukhin et al. [2020] and ANCE Xiong et al. [2021], which mine hard negatives from large corpora. More recently, LLMs have been used to generate domain-specific hard negatives Zhou et al. [2022], Meghwani et al. [2025]. In parallel, contrastive learning for sentence embeddings has shown the value of synthetic hard negatives for robust representations Robinson et al. [2021], Gao et al. [2022]. While these advances highlight the utility of hard negatives, most focus on retrieval or representation learning rather than moderation-specific decision boundaries.

**Synergistic approaches.** Recent work has begun to combine span identification, augmentation, and adversarial example generation. ATGSL Li et al. [2023] rephrases spans to construct adversarial examples, while other pipelines integrate causal span identification with controlled rewriting. However, prior approaches often preserve the causal violation or make coarse sentence-level edits that fail to

Table 1: Comparison with related augmentation approaches

| Method | Span Aware | Label Flip | Policy Guided |
|---|---|---|---|
| Back-translation | No | No | No |
| Token masking | Partial | No | No |
| Counterfactual | No | Yes | No |
| PeerDA | Yes | No | No |
| **Ours** | Yes | Yes | Yes |

guarantee label flips. Our method advances this direction by combining three design elements: (i) agreement-filtered causal span extraction using multiple LLMs, (ii) policy-conditioned rewrites that minimally edit only the violating span, and (iii) validation via reinference to ensure genuine label flips.

This triad distinguishes our framework from prior augmentation methods. By generating boundary-focused hard negatives that are span-sensitive, our approach directly reduces spurious correlations and improves the robustness of moderation models to adversarial manipulations. We summarize how our method compares to related augmentation approaches in Table 1.

## 3 Problem Formulation.

Let dataset $\mathcal{D} = \{(x_i, y_i)\}_{i=1}^{N}$ where $x_i$ is text and $y_i \in \mathcal{Y}$ is its policy label. For violating texts ($y_i \neq$ Compliant), we define:

**Definition 1 (Causal Span):** A substring $s = x_i[a:b]$ is causal for violation $y_i$ if replacing $s$ with policy-compliant text is sufficient to change the label to Compliant.

**Definition 2 (Minimal Causal Set):** The minimal set $S_i = \{s_1, ..., s_k\}$ where replacing all $s_j \in S_i$ flips the label, but replacing any proper subset does not.

Consider the example,

$$x_i = \text{"Win big in our new online casino – join now!"}$$

with $y_i =$ Gambling. A possible set of causal spans is

$$S_i = \{(0, 7), (21, 34)\},$$

corresponding to the substrings "Win big" and "online casino". Replacing these with compliant phrases (e.g., "Enjoy our new board games") would make the ad fall under the Compliant category.

Our objectives are threefold:

1. **Identify:** For a given violating text $x_i$, precisely identify the set of causal spans $S_i$.
2. **Generate:** Produce a new, non-violating text $x_i'$ by editing *only* the content within the identified spans $S_i$, while preserving the non-causal "scaffolding" of the original text.
3. **Fine-tune:** Use the set of newly generated non-violating texts, $\mathcal{A} = \{x_i'\}$, as hard negatives to augment the training data, with the goal of improving the downstream classifier's robustness and reducing its reliance on spurious correlations.

## 4 Method: Span-Aware Counterfactual Augmentation

Our method proceeds in three stages (Figure 1).

- **Span Identification:** Multiple LLMs extract violation-causing spans. An agreement filter ensures high-precision span selection.
- **Span Rewriting:** Identified spans are minimally rewritten into compliant alternatives, guided by policy context.
- **Validation:** Candidate rewrites are re-evaluated; only those that flip the label are retained as hard negatives.

These validated counterfactuals are then injected into classifier training with a tunable augmentation ratio $\alpha$. Unlike traditional augmentation, this process explicitly generates boundary-challenging negatives that sharpen decision boundaries.

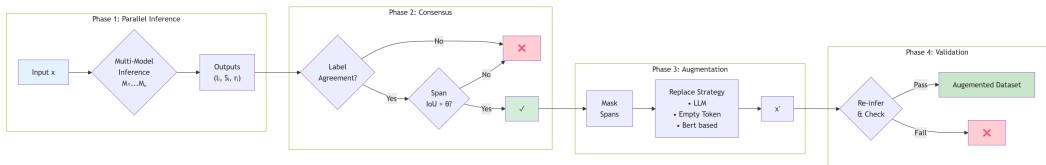

Figure 1: Method Overview showing the four phases: multi-LLM span extraction with agreement filtering, policy-guided span rewriting, re-inference validation, and integration with downstream training.

**Phase 1&2: High-Precision Span Identification.** The pipeline begins with the identification of the causal spans responsible for policy violations. As illustrated in the top-left module of Figure 1, each input ad text is passed to multiple LLMs operating in parallel. Each LLM predicts (i) whether the text violates a policy and (ii) the exact span or spans that trigger the violation. To prevent low-precision or inconsistent annotations from propagating through the pipeline, we introduce an *agreement gate*. This gate enforces two forms of consensus: label agreement (the majority of LLMs must classify the text as violating) and span agreement (the extracted spans must match at the character level across models). Only those samples that pass both criteria are retained. While this stringent filtering reduces the number of usable examples, it ensures that the subsequent stages operate on a seed set of extremely high confidence, minimizing the introduction of noisy or incorrect spans. This choice reflects a precision-over-recall philosophy: better to start with fewer, but cleaner, examples.

**Phase 3: Policy-Guided Span Replacement.** Once high-confidence violating spans have been identified, the next step is to rewrite them into compliant alternatives. This is depicted in the center of Figure 1. We treat the span as the "causal anchor" of the violation, and the rest of the text as scaffolding that should remain untouched. The replacement operator takes as input the original text, the marked span, and the relevant policy rules. It then produces one or more candidate rewrites. Our primary strategy uses an LLM instructed to perform policy-aware rewriting: the model is prompted to replace the violating span with text that is compliant, grammatically coherent, and contextually appropriate for the surrounding sentence. This ensures that the rewritten text is both fluent and realistic, making it a challenging negative example. In addition to this main method, we also implement three baselines: BERT-based token prediction, empty-string removal, and retaining a literal [MASK] token. These alternatives allow us to quantify the importance of contextual, semantic rewriting compared to simpler heuristics. As will be shown in the results section, only the LLM-based rewrites yield meaningful downstream gains.

**Phase 4: Validation and Hard Negative Creation.** A key novelty of our pipeline is that span replacement alone does not guarantee a valid hard negative. Some replacements may fail to change the label, while others may introduce ambiguity or borderline cases. Therefore, as shown in the right-hand block of Figure 1, each candidate rewritten text is re-evaluated by the same set of LLMs used in Phase 1. This secondary inference produces a new consensus label. Only if the majority label is "Compliant" do we consider the rewrite successful, and only then is it added to the pool of hard negatives. This two-step validation ensures that downstream classifiers are trained on examples that are not only syntactically and semantically sound, but also genuinely label-flipped according to the same policy criteria. Importantly, this stage guards against the superficial success of high flip rates from trivial strategies such as empty-string removal: while such methods may achieve flips, they contribute little meaningful signal to the training process. Our validation step filters them out, keeping the focus on informative examples.

**Training Integration.** Finally, Augmented samples can be integrated as hard negatives into the training pipeline of a downstream classifier. To prevent the augmented data from distorting class distributions, we introduce a custom augmentation-aware batch construction as summarized in Algorithm 1:

---

**Algorithm 1** Augmentation-Aware Batch Construction

---

1: **Input:** Original data $\mathcal{D}$, augmented data $\mathcal{A}$, ratio $\alpha$
2: **Output:** Training batch $B$
3: **for** each mini-batch of size $b$ **do**
4:      $n_{aug} \leftarrow \lfloor b \times \alpha \rfloor$
5:      $n_{orig} \leftarrow b - n_{aug}$
6:      Sample $n_{orig}$ from $\mathcal{D}$ preserving class ratios
7:      Sample $n_{aug}$ from $\mathcal{A}$ uniformly
8:      $B \leftarrow \text{shuffle}(D_{orig} \cup A_{aug})$
9: **end for**
10: **return** $B$

---

This approach injects a fixed fraction $\alpha$ of hard negatives into each mini-batch, ensuring a balanced mix of original and augmented samples. By exposing the model to minimally different positive-negative pairs within the same training step, we directly encourage it to learn the fine-grained distinctions that define a policy violation. By training on minimally different pairs of violating and rewritten texts, the classifier is encouraged to focus on the true causal features of a violation, rather than spurious correlations with benign tokens. This design directly addresses one of the most common pitfalls in moderation systems: the tendency to associate superficial patterns (e.g., marketing-related language or common URLs) with violations, leading to high false positive rates. Our span-aware augmentation forces the model to discriminate more carefully, yielding improved robustness in both in-domain and stress-test settings.

## 4.1 Algorithm

The entire pipeline is summarized in Algorithm 2.

---

**Algorithm 2** Span-Aware, Agreement-Filtered Augmentation

---

1: **Input:** Dataset $\mathcal{D}$; LLM set $\mathcal{M}$; policy rules $\pi$; replacement methods $\mathcal{R}$
2: $\mathcal{A} \leftarrow \emptyset$
3: **for** each $x \in \mathcal{D}$ **do**
4:      Query all $m \in \mathcal{M}$ for $(\hat{y}^{(m)}, \hat{S}^{(m)})$
5:      **if** label and span agreement across $\mathcal{M}$ **then**
6:          Mask $\hat{S}$ in $x$ to obtain $x^{\text{mask}}$
7:          **for** each method $\in \mathcal{R}$ **do**
8:              $x' \leftarrow \text{replace}(x^{\text{mask}}, \hat{S}, \pi, \text{method})$
9:              $\hat{y}' \leftarrow \text{majority}\big(\text{infer}(x', m)_{m \in \mathcal{M}}\big)$
10:              **if** $\hat{y}' = \textit{Compliant}$ **then**
11:                  $\mathcal{A} \leftarrow \mathcal{A} \cup x'$
12:              **end if**
13:          **end for**
14:      **end if**
15: **end for**
16: **return** $\mathcal{A}$

---

**System-Level Perspective.** While Algorithm 1 provides a procedural pseudocode for our method, the architecture diagram (Figure 1) highlights the system-level flow and clarifies how the components interact. Inputs move left to right: from raw violating texts, through the LLM ensemble for span extraction, into the rewriting module, back through validation, and finally into the training data pipeline. This modular design makes the system extensible, new replacement strategies or agreement mechanisms can be plugged in without altering the overall flow.

**Summary.** In short, the architecture enforces three principles: (i) *precision-first filtering*, ensuring only high-confidence seeds enter the pipeline, (ii) *contextual rewriting*, which produces realistic and challenging negatives, and (iii) *validation before integration*, which prevents noise from contaminating

training. Together, these design choices distinguish our pipeline from simpler augmentation methods and underpin the robustness improvements we observe in downstream experiments.

# 5 Experimental Setup

**Datasets** Our experiments leverage three distinct datasets to evaluate our pipeline's performance, robustness, and generalizability.

- **Internal Adult Content Dataset:** Our primary dataset is a real-world collection of ad texts related to adult content policies, comprising 16,000 training, 4,000 validation, and 3,000 gold-standard test samples. The dataset is characterized by its subtlety and the nuanced language used to express policy violations.

- **Kaggle Toxic Comment Dataset:** To assess generalization to a public, multi-label benchmark, we use the Kaggle Toxic Comment Classification Challenge dataset. It includes six labels: toxic, severe toxic, obscene, threat, insult, and identity hate. To simulate a more realistic low-resource scenario, we subsampled the original 159.5K training samples down to 18K training and 2K validation instances, while keeping the full test set of 63.9K samples for a comprehensive evaluation.

- **"Spy-Camera" Stress Set:** To specifically measure robustness against spurious correlations, we constructed a diagnostic stress set. This set contains examples featuring common but benign trigger words (e.g., "spy camera") where any genuinely violating content has been neutralized. A model that overfits to such trigger words will perform poorly on this set, allowing us to directly quantify the improvements in robustness.

Details of the dataset distribution is presented in Table 2

Table 2: Dataset Summary. Please Note: Label Wise Details present the samples for each label in the kaggle dataset

| Dataset | Compliant | Violating | Label Wise Details |
|---|---|---|---|
| Internal - Training | 8000 | 8000 | |
| Internal - Validation | 2000 | 2000 | |
| Internal - Testing | 2743 | 321 | |
| Kaggle - Training | 7208 | 792 | 756,82,428,20,384,64 |
| Kaggle - Validation | 1803 | 197 | 188,19,106,4,95,17 |
| Kaggle - Testing | 57735 | 6243 | 6090,367,3691,211,3427,712 |
| Stress Test - Testing | 250 | 150 | |

**Models and Implementation** For the augmentation pipeline, we utilize **GPT-4o** and **DeepSeekV3** for all LLM-dependent stages: initial span extraction, policy-guided rewriting, and re-inference validation. The downstream classifier targeted for improvement is a **BERT-base** model Devlin et al. [2019] specifically, we use the bert-base-uncased variant, which has 110 million parameters. We fine-tune the classifier for a maximum of 15 epochs using the AdamW optimizer with a learning rate of 2e-4 and a batch size of 128. Early stopping is employed based on validation set performance to prevent overfitting. All reported downstream results are averaged over 5 runs with different random seeds to ensure statistical significance.

**Evaluation Metrics and Baselines** We evaluate the effectiveness of our replacement strategies using the Flip Rate, defined as the percentage of violating samples successfully converted to a compliant label after augmentation and re-inference. For the downstream classification task, our primary metric is the Area Under the Precision-Recall Curve (PRAUC), which is well-suited for the class imbalance often present in content moderation tasks. For the multi-label Kaggle dataset, we report the macro-averaged PRAUC across all classes.

Our primary baseline is the BERT classifier fine-tuned with no data augmentation ($\alpha$=0%). We also compare against a common heuristic baseline, **Random Span Masking**, where a random span of text, with length sampled from the distribution of true violating spans are masked. This comparison allows us to isolate the benefits of our targeted, span-aware approach from the general effects of masking.

**Computational Cost:** Average 0.3 seconds per span extraction, 0.5 seconds per rewrite (GPT-4o API). Total augmentation pipeline: $\sim$4 hours for 10K samples.

# 6   Experimental Results

**Data Quality from the Agreement Gate.**   We first evaluated the effectiveness of our agreement gate on a subset of violating ad texts from our internal dataset. Of these, 90.3% achieved label agreement between our two LLMs, and 61.2% achieved exact joint span agreement. This demonstrates that the agreement gate acts as a strong precision-oriented filter, producing a compact, high-confidence seed set for augmentation.

**Quality of Span Replacements.**   We next evaluated span replacement strategies on these high-agreement samples. As shown in Table 3, the policy-guided LLM rewrite achieved the highest flip rate (94.3%), outperforming BERT-based token replacement (90.7%), empty-string removal (90.4%), and leaving masked spans unchanged (88.1%). While simpler strategies achieve reasonably high flip rates, they often strip away useful context, limiting their downstream value. In contrast, contextual, policy-aware rewrites preserve sentence structure and semantics, yielding realistic hard negatives that are more informative for training.

Table 3: Flip rate after span replacement on the high-confidence subset. Contextual LLM rewrites achieve the highest success rate.

| Replacement Method | Flip % |
|---|---|
| **LLM-based (policy-guided)** | **94.3%** |
| BERT token replacement | 90.7% |
| Empty-string removal | 90.4% |
| Keep [MASK] | 88.1% |

**Main Downstream Impact on Internal Dataset.**   The primary test of our method is whether generated hard negatives improve downstream classification. On the internal adult test set, augmenting with span-aware negatives yields consistent PRAUC improvements (Figure 2). Performance rises from a baseline of 0.73 to 0.79 at an augmentation fraction ($\alpha$) of 20%. This shows a substantial 6.3% relative improvement in PRAUC. Interestingly, after this peak, we observe a slight decline in performance. This suggests a critical trade-off: while hard negatives are highly informative, adding too many can oversaturate the training batches with compliant examples. This can dilute the signal from the original, scarcer positive samples, leading the model to become overly cautious. This finding highlights the importance of tuning the augmentation ratio, $\alpha$, as a key hyperparameter.

In contrast, the Random Span Masking baseline (Table 4) severely degrades performance, dropping from 0.73 to 0.50 as augmentation increases. This confirms that performance gains stem not from augmentation per se, but from the targeted, semantics-preserving nature of our edits.

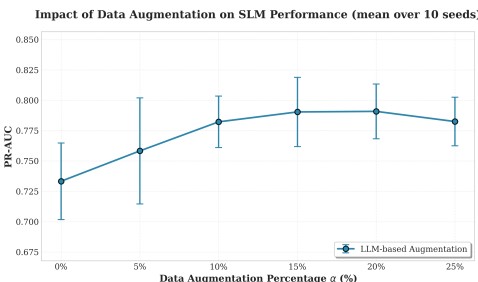

Figure 2: Downstream performance on the internal adult test set. PRAUC improves with span-aware augmentation, peaking at 20%. Error bar indicate Std across seeds/

**Ablation: The Role of Contextual Rewriting.**   To isolate the source of these gains, we compared LLM rewrites with simpler strategies such as empty-string removal and BERT token replacement. As shown in Figure 3, both alternatives degrade PRAUC sharply, even at small augmentation fractions. This underscores a key finding: high flip rates alone are insufficient. Without preserving context and fluency, replacements confuse the classifier rather than strengthen it. Contextual rewriting is therefore

Table 4: PRAUC vs. augmentation fraction $\alpha$ on the internal adult test set (mean of 5 seeds).

| Method | 0% | 5% | 10% | 15% |
|---|---|---|---|---|
| Span-aware aug. | 73.3% | 75.8% | 78.2% | 79.0% |
| Random span mask. | 73.3% | 65.9% | 58.5% | 50.1% |

critical for producing informative hard negatives. This is particularly true for simpler strategies like BERT token replacement, where the resulting text often becomes too short and contextually uninformative to serve as a useful training example.

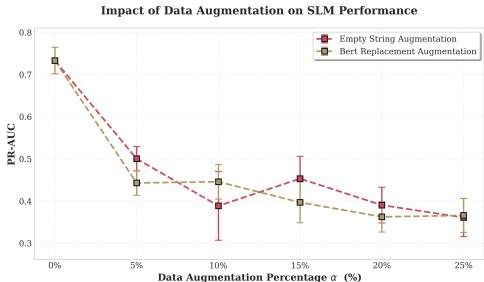

Figure 3: Ablation results. Simpler strategies degrade PRAUC, highlighting the necessity of contextual LLM rewrites.

**Generalization and Robustness.** We next tested robustness and generalization. On the "spy-camera" stress set, designed to expose spurious correlations, the baseline achieved PRAUC ~0.634, while span-aware augmentation improved performance to ~0.765 with 20% injection. This confirms that our approach reduces reliance on trigger words and better captures true violation signals.

We also applied the pipeline to the Kaggle Toxic Comment dataset, a multi-label classification task. As shown in Figure 4, macro-PRAUC improved across classes, peaking at an augmentation fraction of ~5%. The optimal augmentation ratio is notably lower here than for our internal dataset (5% vs. 20%). This is a significant observation. It suggests that for broader, more diverse violation types like those in the Kaggle dataset, the decision boundary is more complex. A smaller injection of hard negatives is sufficient to refine the boundary without distorting the model's understanding of the varied vocabulary and phrasing associated with toxicity. In contrast, the more formulaic nature of ad policy violations may allow for a higher proportion of hard negatives before the signal from positive samples is overwhelmed.

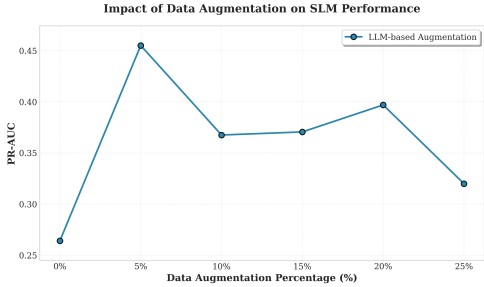

Figure 4: Performance on the Kaggle multi-label toxic comment dataset. Improvements peak at ~5% augmentation.

## 6.1 Error Analysis

Despite the high flip rate, a small number of samples remain unflipped. These errors fall into two categories: (i) **incomplete span extraction**, often involving complex URLs, emojis, or multi-span

violations, and (ii) **policy edge cases**, where rewrites remain borderline compliant and are still flagged by re-inference.

## 6.2 Discussion

Our findings underscore a central principle for reliable ML under unreliable data: it is preferable to generate fewer, high-quality counterfactuals than to flood training with noisy or label-preserving augmentations. Agreement filtering and validation enforce this precision-first approach.

The observed **task-dependent augmentation** ratios further highlight that robustness is not a one-size-fits-all property. Instead, reliability emerges from balancing adversarial counterexamples against original positive signal. This perspective reframes augmentation not as "more data = better," but as a controlled intervention that exposes vulnerabilities while preserving generalization.

## 7 Limitations and Future Work

While our span-aware framework advances robustness, it is not without limitations. The pipeline depends on LLM judgments for span extraction, rewriting, and validation; although the agreement gate reduces errors, residual unreliability remains, particularly for multi-span violations, URLs, emojis, and obfuscated text. Since rewrites are drawn from known categories of violations, emerging or novel adversarial tactics (e.g., misinformation, scams) may be underrepresented. Moreover, reliance on proprietary LLM APIs introduces costs, scalability constraints, and potential vendor lock-in.

Future work will explore integrating open-weight LLMs, semantic span matching for obfuscated violations, and broader stress tests to evaluate resilience under distribution shift. More generally, the pipeline points toward a modular and auditable approach to counterfactual generation for reliable ML, one that is applicable well beyond content moderation to other domains where unreliable and strategically manipulated data threaten robustness.

### 7.1 Discussion

Our results show that building robust models from strategically manipulated data requires both precision in identifying causal signals and realism in generating counterfactuals. The 61.2% span agreement rate, for instance, is not a limitation but a feature: it reflects a deliberate precision-over-recall tradeoff. In the context of unreliable data, it is better to generate fewer, high-quality augmentations than to introduce noisy, low-quality data that could further destabilize training.

Furthermore, the discovery of task-dependent optimal augmentation ratios (5-20%) is a key practical insight for building reliable systems. It suggests that practitioners should treat the augmentation ratio $\alpha$ as a tuning parameter dependent on the complexity and adversarial nature of their specific domain. This reinforces a core principle of reliable ML: there is a fundamental tradeoff between hardening a model against known manipulation patterns and preserving its generalization to broader, benign data. Too little augmentation fails to break spurious correlations; too much overwhelms the genuine signals from positive examples.

## 8 Conclusion

We introduced a span-aware, agreement-filtered augmentation pipeline to address the critical challenge of training reliable classifiers from strategically manipulated text. Our framework generates high-quality, label-flipping hard negatives that force models to learn the true causal signals of a policy violation, rather than relying on spurious correlations. Our experiments demonstrate that: (1) contextual, policy-guided rewrites reliably generate valid counterfactuals from violating examples, (2) strategically injecting these hard negatives (5-20%) significantly improves downstream model robustness and performance on real-world moderation tasks and (3) this approach makes classifiers more robust to trigger words and generalizes across different domains of manipulated content, from ads to toxic comments. This work provides a practical, deployable path toward building more robust and reliable content moderation systems, a crucial step in fostering safer online environments.

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
