# OpenReview forum: "Disarming Strategic Text: Span-Aware Counterfactuals for Robust Content Moderation"
_NeurIPS.cc/2025/Workshop/Reliable_ML — NeurIPS 2025 - Reliable ML Workshop_

### Official Review · Reviewer_LJPi · 2025-09-12
**Careful and Comprehensive Pipeline with Significant Performance Gains**

**Rating:** 6
**Confidence:** 4

**Review:**

The paper proposes a pipeline that incorporates three components (span-aware modeling, label-flip handling, and policy-guided augmentation) to generate counterfactual examples. The results demonstrate both performance improvements and the trade-offs associated with applying these techniques in different proportions.

However, one concern is that the reported impact of augmentation appears surprisingly large compared to the baseline with 0% augmentation (alpha). For instance, Figure 2 shows an improvement of nearly 6% at alpha = 20%, while Figure 3 reports a 20% increase at alpha = 5%. This raises questions about what drives such substantial gains. Additional analysis of the augmented examples would be valuable to assess whether these improvements are meaningful and interpretable from a human perspective.

---

### Official Review · Reviewer_YVTV · 2025-09-19
**Good fit for the workshop, with clear potential for improvement, especially in presentation, detail, and comparative evaluation**

**Rating:** 8
**Confidence:** 3

**Review:**

# *Summary*

The paper proposes a data augmentation pipeline for improving the robustness of content moderation models against strategic text manipulations. They first identified violation-causing spans by prompting multiple LLMs and keeping only cases where they agree on both the label and the exact span. These spans are then rewritten into compliant alternatives using LLMs. Finally, the rewritten texts are re-evaluated by the same LLM ensemble, and only those that genuinely flip the label to “Compliant” are retained as counterfactuals. The newly augmented dataset was tested and results were shared, with PRAUC improvement up to \+6.3 points on internal ad dataset.

# *Strengths.*

- The approach uses multiple LLM agreement filtering and re-inference validation to ensure quality.

- Evaluated on multiple datasets, multiple seeds, and results included ablations.

- Very relevant to the workshop, directly addresses reliability under imperfect and adversarial data.

- Clear definitions, and explanations for why certain decisions were made. Their philosophy to the approach is very clear to the reader.

# *Weaknesses / Limitations.*

* “Policy guided”

  * Please elaborate on what “policy-guided” means. Which policies were used?

  * Share the prompts given to the LLMs. Did the LLM identify the violated policy explicitly, or just mark text as violating and highlight spans?

  * Was any human validation performed on a subset of LLM outputs?

* “Spy camera stress set”: I would like more examples showing how the “Spy camera stress set” differs from the other datasets. You mention it is designed to expose spurious correlations, please elaborate how.

* A comparison with what is considered sota in sentence level edits for generating counterfactual examples, compared with your work is important. I*s your approach to span level edit truly making datasets more robust than existing sentence-level ones?* Or are we looking at a comparable unique approach that works just as well?

* How were euphemisms taken care of? You mention them in abstract and conclusion, , but the mechanism is unclear. Please explain how they were addressed.

* Examples of false negatives that escaped your method would be valuable. A qualitative analysis could highlight gaps and guide future work.

# *Suggestions for Authors.*

* Provide examples early. Show original text alongside augmented versions generated ideally page 1 or 2\.

* Figures: the text is very small and hard to read. Reworking them would make them more accessible.

* Accessibility: scientific writing should avoid decorative language. For example, line 134-135 says

  > “We treat the span as the ‘causal anchor’ of the violation, and the rest of the text as scaffolding that should remain untouched.”

  This could be simplified while still keeping precise vocabulary.

* Table 2 could be simplified (e.g., one row for each dataset , with more columns to show details for train, val or test). A bar chart might help more than a long list of numbers for the label wise details. What are the labels?.

* Results in a table with all the different approaches, PRAUC before and after, at each augmentation level , absolute difference between before and after, is the difference statistically significant etc will really help with understanding your work better and easier.

* Combining Figures 2 and 3 to make comparisons between strategies easier to follow.

* For computational cost, I am seeing more papers that also add the monetary cost ( mostly in $) , so that the reader doesn’t have to do the math.